# Identification of Novel HBV/HDV Entry Inhibitors by Pharmacophore- and QSAR-Guided Virtual Screening

**DOI:** 10.3390/v13081489

**Published:** 2021-07-29

**Authors:** Michael Kirstgen, Simon Franz Müller, Kira Alessandra Alicia Theresa Lowjaga, Nora Goldmann, Felix Lehmann, Sami Alakurtti, Jari Yli-Kauhaluoma, Karl-Heinz Baringhaus, Reimar Krieg, Dieter Glebe, Joachim Geyer

**Affiliations:** 1Institute of Pharmacology and Toxicology, Faculty of Veterinary Medicine, Justus Liebig University Giessen, 35392 Giessen, Germany; michael.kirstgen@vetmed.uni-giessen.de (M.K.); Simon.Mueller@vetmed.uni-giessen.de (S.F.M.); Kira.A.Lowjaga@vetmed.uni-giessen.de (K.A.A.T.L.); 2Institute of Medical Virology, National Reference Center for Hepatitis B Viruses and Hepatitis D Viruses, Justus Liebig University Giessen, 35392 Giessen, Germany; Nora.Goldmann@viro.med.uni-giessen.de (N.G.); Felix.Lehmann@viro.med.uni-giessen.de (F.L.); Dieter.Glebe@viro.med.uni-giessen.de (D.G.); 3Drug Research Program, Division of Pharmaceutical Chemistry and Technology, Faculty of Pharmacy, University of Helsinki, Viikinkaari 5 E, FI-00014 Helsinki, Finland; sami.alakurtti@neste.com (S.A.); jari.yli-kauhaluoma@helsinki.fi (J.Y.-K.); 4VTT Technical Research Centre of Finland, Biologinkuja 7, FI-02044 Espoo, Finland; 5Sanofi-Aventis Deutschland GmbH, 65926 Frankfurt, Germany; Karl-Heinz.Baringhaus@sanofi.com; 6Institute of Anatomy II, University Hospital Jena, Teichgraben 7, 07743 Jena, Germany; REIMAR.KRIEG@med.uni-jena.de; 7German Center for Infection Research (DZIF), Partner Site Giessen-Marburg-Langen, 35392 Giessen, Germany

**Keywords:** HBV, HDV, NTCP, entry inhibitor, QSAR, pharmacophore, virtual screen

## Abstract

The hepatic bile acid transporter Na^+^/taurocholate co-transporting polypeptide (NTCP) was identified in 2012 as the high-affinity hepatic receptor for the hepatitis B and D viruses (HBV/HDV). Since then, this carrier has emerged as promising drug target for HBV/HDV virus entry inhibitors, but the synthetic peptide Hepcludex^®^ of high molecular weight is the only approved HDV entry inhibitor so far. The present study aimed to identify small molecules as novel NTCP inhibitors with anti-viral activity. A ligand-based bioinformatic approach was used to generate and validate appropriate pharmacophore and QSAR (quantitative structure–activity relationship) models. Half-maximal inhibitory concentrations (IC_50_) for binding inhibition of the HBV/HDV-derived preS1 peptide (as surrogate parameter for virus binding to NTCP) were determined in NTCP-expressing HEK293 cells for 150 compounds of different chemical classes. IC_50_ values ranged from 2 µM up to >1000 µM. The generated pharmacophore and QSAR models were used for virtual screening of drug-like chemicals from the ZINC^15^ database (~11 million compounds). The 20 best-performing compounds were then experimentally tested for preS1-peptide binding inhibition in NTCP-HEK293 cells. Among them, four compounds were active and revealed experimental IC_50_ values for preS1-peptide binding inhibition of 9, 19, 20, and 35 µM, which were comparable to the QSAR-based predictions. All these compounds also significantly inhibited in vitro HDV infection of NTCP-HepG2 cells, without showing any cytotoxicity. The best-performing compound in all assays was ZINC000253533654. In conclusion, the present study demonstrates that virtual compound screening based on NTCP-specific pharmacophore and QSAR models can predict novel active hit compounds for the development of HBV/HDV entry inhibitors.

## 1. Introduction

Chronic hepatitis following hepatitis B (HBV) and D (HDV) virus infections is the main cause of hepatocellular carcinoma (HCC) and liver cirrhosis. Even the availability of vaccination does not prevent the more than 800,000 deaths annually from the long-term effects of chronic liver inflammation associated with HBV/HDV infections [1]. Both viruses are coated with identical envelope proteins that are coded by the 3.2 kb DNA genome of HBV [2]. The RNA genome of HDV does not code for any envelope protein. Therefore, HDV acts as a so-called satellite virus and makes use of the envelope proteins derived from the HBV genome [3,4]. This is the reason why HDV can only spread in the presence of an HBV infection. In addition, the interaction of both viruses with their cellular entry receptor NTCP is based on the common envelope proteins (HBs) [5]. NTCP (Na^+^/taurocholate co-transporting polypeptide, gene symbol *SLC10A1*) represents a physiologically important hepatic bile acid transporter and was also identified as the hepatic entry receptor for HBV and HDV. High affinity binding of both viruses to NTCP is mediated by the myristoylated preS1 domain (myr-preS1_2–48_ lipopeptide), consisting of the 2–48 N-terminal amino acids of the large surface protein (LHBs). As a common but mostly non-curative therapy for virus-related chronic hepatitis, nucleos(t)ide-analogue (NUC) reverse transcriptase inhibitors (for HBV) and interferon (for both HBV/HDV) are used. Unfortunately, interferon therapy is highly prone to adverse drug reactions and NUCs have to be given life-long [6,7].

Identification of NTCP as an hepatic receptor for HBV/HDV in 2012 enabled the development of NTCP inhibitors as HBV/HDV entry inhibitors [8,9]. The mentioned myr-preS1_2–48_ lipopeptide itself has the ability to block in vitro HBV/HDV infection with inhibitory constants (IC_50_) in the low nanomolar range [4]. Based on that, a synthetic analogue of this lipopeptide (Hepcludex^®^) has been developed and was recently approved as first HDV entry inhibitor interacting with NTCP [10]. Furthermore, numerous studies identified novel chemical entry inhibitors for HBV and HDV by screening for bile salt transport inhibitors [11] or by screening for inhibitors of myr-preS1_2–48_ lipopeptide attachment and/or in vitro HBV/HDV infection [12] in appropriate hepatoma cell culture models overexpressing NTCP. However, to date, none of these small molecules have been able to enter the clinical development phase so far.

In previous studies, we identified individual compounds from two different classes (betulin and propanolamine derivatives) that were quite potent for myr-preS1_2–48_ lipopeptide binding inhibition and significantly blocked in vitro HDV infection of NTCP-expressing HepG2 cells [13,14]. In the present study, we aimed to expand the group of small molecule NTCP inhibitors by pharmacophore-based virtual screening (VS) of compound libraries.

A well-established method of identifying novel inhibitor candidates for cellular drug targets is high-throughput screening (HTS) of chemical libraries. This technique can rapidly generate data of large subsets of molecules using automated experimental assays [15]. However, hit rates of HTS are only between 0.01% and 0.1% [16], which leads to immense drug discovery costs [17]. An alternative is given by so-called quantitative structure–activity relationship (QSAR) analysis as a ligand-based method for drug design [18]. In principle, this represents a bioinformatic method for building mathematical models that describe the correlation between physicochemical properties of ligands and continuous (IC_50_, EC_50_, *K*_i_, etc.) or categorical (active, inactive, toxic, nontoxic, etc.) properties, by using statistical regression techniques [19]. Nowadays, these models are used for VS approaches to predicting activities for compounds of large chemical databases (e.g., ZINC^15^ [20]). Only substances with the best predicted activities are then selected for experimental validation [21]. Hit rates of this method range from 1% to 40% depending on the predictability of the generated model [22]. Compared to experimental HTS, QSAR-based VS of chemical libraries often results in higher hit rates of biologically active compounds at lower costs [22]. In the present study, we demonstrate that by QSAR-based VS novel small molecule inhibitors of NTCP can be identified that indeed showed proof-of-concept inhibition of HDV infection.

## 2. Materials and Methods

### 2.1. NTCP-Expressing Cell Lines

Human embryonic kidney (HEK293) cells were stably transfected with human NTCP and C-terminally tagged with the FLAG epitope (further referred to as NTCP-HEK293 cells) as reported before [23,24]. Cells were maintained at 37 °C, 5% CO_2_, and 95% humidity in DMEM/F-12 medium (Thermo Fisher Scientific, Waltham, MA, USA) supplemented with 10% fetal calf serum (Sigma-Aldrich, St. Louis, MO, USA), 4 mM L-glutamine (PAA, Cölbe, Germany), and penicillin/streptomycin (PAA). HepG2 cells stably transfected with NTCP-FLAG (further referred to as NTCP-HepG2 [9]) were cultured under the same conditions in DMEM with all supplements listed above, except for L-glutamine. For induction of the transgene, the medium was supplemented with 1 µg/mL tetracycline (Roth, Karlsruhe, Germany) in the case of the NTCP-HEK293 cells or with 2 µg/mL doxycycline (Sigma-Aldrich) in the case of the NTCP-HepG2 cells.

### 2.2. Inhibitory Concentrations (IC_50_) for [^3^H]preS1 Binding and [^3^H]TC Transport

Bile acid transport measurements were performed in NTCP-HEK293 cells with tritium-labelled taurocholic acid (further referred to as [^3^H]TC) (20 Ci/mmol, 0.09 mCi/mL, Perkin Elmer, Waltham, MA, USA). In parallel, peptide-binding experiments were performed with a tritium-labelled myr-preS1_2–48_ lipopeptide -HBV subgenotype D3- (further referred to as [^3^H]preS1) that was purchased from Pharmaron (120 Ci/mmol, 1 mCi/mL, Cardiff, UK) as reported [13]. Briefly, cells were seeded onto polylysine-coated 96-well plates, induced with 1 µg tetracycline per ml, and grown to confluence over 72 h at 37 °C. Then, cells were washed once with tempered phosphate buffered saline (PBS, 137 mM NaCl, 2.7 mM KCl, 1.5 mM KH_2_PO_4_, 7.3 mM Na_2_HPO_4_, pH 7.4) at 37 °C and preincubated with 80 µL DMEM for 5 min at 37 °C. The medium was replaced by 80 µL DMEM containing the respective inhibitor (concentrations ranging from 0.1 to 1000 µM) or solvent alone (100% uptake/binding control), and cells were further incubated for 5 min at 37 °C. After this pre-incubation, bile acid transport experiments were started by adding 20 µL DMEM containing 5 µM [^3^H]TC (final concentration: 1 µM). Binding of [^3^H]preS1 was initiated by adding 20 µL DMEM containing 25 nM [^3^H]preS1 (final concentration: 5 nM). Experiments were stopped after 10 min by washing twice with ice-cold PBS. For 0% uptake/binding control, the NTCP-HEK293 cells were not induced with tetracycline (-tet). Cell-associated radioactivity of [^3^H]TC or [^3^H]preS1 was quantified by liquid scintillation counting in a Packard Microplate Scintillation Counter TopCount NXT (Packard Instrument Company, Meriden, USA). Transport rates and [^3^H]preS1 binding were determined in counts per minute (cpm). The mean of the 0% control was subtracted and the net [^3^H]TC transport rates and net [^3^H]preS1 binding rates, respectively, were expressed as % of control. IC_50_ values were calculated from quadruplicate determinations by GraphPad Prism 6 (GraphPad, San Diego, CA, USA).

### 2.3. HDV Infection Experiments

HDV production was done in vitro as described before [25,26]. RT-qPCR was performed to determine HDV RNA genome equivalents. NTCP-HepG2 cells were pre-incubated for 5 min with inhibitors solved in 80 µL hepatocyte growth medium (HGM) per well in concentrations ranging from 5 µM to 300 µM. Infection experiments were performed in NTCP-HepG2 cells as described [13]. Briefly, during infection, cells were cultured in 96-well plates in HGM consisting of William’s E Medium (Thermo Fisher Scientific) containing 2% bovine serum albumin (BSA, Roth), 2 mM L-glutamine (Thermo Fisher Scientific), 100 µg/mL gentamicin (Thermo Fisher Scientific), 10 nM dexamethasone (Sigma-Aldrich), 1 mM sodium pyruvate (Thermo Fisher Scientific), 1× insulin-transferrin-selen (Thermo Fisher Scientific), 2% DMSO (Merck, Darmstadt, Germany), 4% polyethylene glycol (Sigma-Aldrich), and 2 µg/mL doxycycline (Sigma-Aldrich). HDV stock solved in 20 µL HGM per well was added for infection and cells were incubated for 6 h with final concentration of 120 genome equivalents/cell of HDV particles. Subsequently, cells were washed with DMEM and cultured in HGM supplemented with 2% DMSO, 2% BSA, and 2 µg/mL doxycycline. Every three days medium was changed until cells were fixed at day 9 post infection with 3% paraformaldehyde (Sigma-Aldrich) in PBS, for 30 min at room temperature (RT). Cells were permeabilized with 0.2% Triton X 100 (Roth) in PBS for 30 min at RT, and blocked by incubation with 5% bovine serum albumin (Roth) in PBS, for 30 min at RT. Then, cells were immunostained with purified human anti-HDV-positive serum at 37 °C for 1 h (1:400 dilution). Goat anti-human IgG secondary antibody coupled to Alexa Fluor fluorophore (1:400 dilution, Thermo Fisher Scientific) was added for 1 h at 37 °C for detection of hepatitis delta antigen (HDAg) as described before [27]. Nuclei were stained with Hoechst 33,342 (1 µg/mL, Thermo Fisher Scientific).

### 2.4. Cytotoxicity Assay

The In Vitro Toxicology Assay Kit (Sigma-Aldrich) was used to perform a 3-[4,5-dimethylthiazole-2-yl]-2,5-diphenyltetrazolium bromide (MTT) assay to measure the cytotoxicity of the indicated compounds according to the manufacturer’s protocol. Briefly, NTCP-HepG2 cells were incubated with 100 µL of the indicated concentrations of the respective compound solved in HGM over 6 h at 37 °C. After 6 h, the medium was replaced by inhibitor-free HGM and cells were cultured for additional 24 h. Then medium was removed and 100 µL DMEM containing 0.5 mg/mL MTT were added and cells were incubated for 1 h at 37 °C. Finally, the medium was replaced by 100 µL isopropyl alcohol (Sigma-Aldrich) and samples were measured by ELISA reader (GloMax-Multi Detection System, Promega, Madison, WI, USA).

### 2.5. Tested Compounds

A set of betulin derivatives (TargoSet) was purchased from Adipogen AG (Liestal, Switzerland). All other betulin derivatives were synthesized as outlined elsewhere [28,29,30]. In total, data from 31 betulin derivatives were taken into account. A set of 87 test compounds was provided by Sanofi-Aventis Deutschland GmbH (Frankfurt, Germany) [14]. A set of 18 arylmethylamino steroids was provided by Krieg et al. [31]. A set of 20 compounds, which can be found in the ZINC^15^ database [20] (https://zinc.docking.org/, 7 May 2021), was purchased from MolPort (Riga, Latvia). All other tested compounds were purchased from Sigma-Aldrich.

### 2.6. Data Preparation

The software MAESTRO Molecular Modeling Interface (Version 12.2) of SCHRÖDINGER, Inc. (https://www.schrodinger.com/, 15 June 2020, New York City, NY, USA) was used. All 2D structures were imported as *sd*files into the MAESTRO Molecular Modeling Interface and were cured and standardized using LigPrep [32] with the following settings: force field OPLS_2005 (default setting); ionization state at target pH 7.4 (Epik; custom setting); desalt (default setting); generate tautomers (default setting); retain specified chiralities (default setting).

### 2.7. Pharmacophore Generation

Generation of pharmacophore model was performed using PHASE [33] with the following settings: active = IC_50_ < 10 µM (binding affinity > 5; custom setting); inactive = IC_50_ > 20 µM (binding affinity < 4.7; custom setting); hypothesis should match at least 50% of actives (default setting); 4 to 5 features in the hypothesis (default setting); difference criterion 0.5 (default setting); create excluded volume shell from actives and inactives (default setting); minimum number of inactives that must experience a clash = 1 (default setting); minimum distance between active surface and excluded volumes 1 Å (default setting); excluded volume sphere radii 1 Å (default setting).

### 2.8. Atom-Based QSAR-Model Generation

Atom-based QSAR models were built using PHASE [33] with the following settings: random training set 70% (custom setting) (59 compounds training set, 26 compounds test set); grid spacing = 1 Å (default setting); maximum PLS factors = 7 (custom setting). Determined IC_50_ values were used as property input, expressed as “binding affinity” = −log (IC_50_ [M]).

### 2.9. Virtual Screening

The downloaded database was prepared for pharmacophore-screening using PHASE [33] with the following settings: skip duplicate ligands (default setting); generate 10 ligand conformers and minimize output (custom setting); generate possible states at pH 7.4 (epik; custom setting); retain specified chiralities (default setting); retain at most 4 low-energy stereoisomers per ligand (default setting); generate up to 1 low-energy 5- and 6-membered ring conformations (default setting); remove high-energy ionization/tautomer states (default setting); prefilter by Lipinski’s Rule (custom setting).

### 2.10. Statistics

Determination of IC_50_ values was done by nonlinear regression analysis using the equation log(inhibitor) vs. response settings of the GraphPad Prism 6.0 software (GraphPad). Data of bile acid transport and [^3^H]preS1 binding were expressed as means ± SD from quadruplicate determinations. Infection studies show data from three independent experiments, each with triplicate determinations represented as means ± SD. Statistical analysis of the HDV infection experiments were performed by two-way ANOVA, followed by Dunnett’s multiple comparison test by GraphPad Prism 6.0, considering *p* < 0.01 as statistically significant. Statistical data of presented bioinformatic models originated from the MAESTRO Molecular Modeling Interface.

## 3. Results

The most important factor for the predictability of pharmacophore and QSAR models is the quality of data input [34]. To ensure input of data from congeneric experiments, we performed all experiments with the identical cell line and protocol, as described before [13,14]. Namely, we used all test compounds as inhibitors for [^3^H]preS1-peptide binding inhibition to NTCP and [^3^H]TC transport inhibition via NTCP, both in NTCP-expressing HEK293 cells. These experiments were performed with four different groups of test compounds, including 31 betulin-derived triterpenoids [13], and 87 propanolamine derivatives [14]. Both groups of compounds were analyzed before as novel NTCP inhibitors in our laboratory. In addition, a set of 18 arylmethylamino steroids that previously showed antiparasitic activity against *Plasmodium falciparum* and *Schistosoma mansoni* [31] was used for NTCP inhibition for the first time in the present study (Appendix A). And finally, a group of 55 structurally unrelated compounds that were reported as NTCP inhibitors in the literature [35,36,37,38,39,40,41,42,43,44,45,46] were included in the [^3^H]preS1-peptide binding inhibition experiments. Figure 1 gives an overview about the workflow of the present study.

In a first step, all compounds were used at 100 µM inhibitory concentration in binding experiments with 5 nM [^3^H]preS1-peptide and transport experiments with 1 µM [^3^H]TC in NTCP-overexpressing HEK293 cells (Appendix A). For all compounds that showed more than 50% inhibition of [^3^H]preS1-peptide binding inhibition, detailed half-maximal inhibitory concentrations (IC_50_) were determined. However, for some of the compounds a sigmoidal concentration-dependent dose-response relationship could not be determined experimentally. Therefore, these compounds were removed from the dataset. This was important to avoid distortion of subsequent models due to non-valid IC_50_ data. In total, a data set consisting of 85 compounds with valid IC_50_ data could be compiled. IC_50_ values for the betulin [13] and propanolamine [14] derivatives were taken from previous studies of our laboratory. The inhibition pattern of the arylmethylamino steroids is presented in detail in Appendix A. The IC_50_ values for [^3^H]preS1-peptide binding inhibition of this compound class ranged from 8–251 µM. The most potent compound was steroid 12c with an IC_50_ for [^3^H]preS1-peptide binding inhibition of 8 µM. Structures of the three most potent [^3^H]preS1-peptide binding inhibitors of this compound class are depicted in Appendix A.

For QSAR modeling, all IC_50_ values were transformed by −log(IC_50_ [M]) conversion into binding affinity values as indicated in Table 1 In addition, data of the chemical structures were collected and were saved as 2D *sd*files. For curation and standardization of the chemical structures, LigPrep [32] was used (encoded in the software of the Schrödinger Suite which can be executed through the Maestro graphical user interface). In total, the IC_50_ values for [^3^H]preS1-peptide binding inhibition from the 85 data set compounds revealed binding affinities between 1 (IC_50_ > 1000 µM) and5.699 (IC_50_ = 2 µM). The compounds then were divided into a training set to generate the QSAR model (59 compounds, Table 1) and a test set to validate the QSAR model (26 compounds, Table 4) as described in the Material and Methods Section 2.8.

The atom-based QSAR model, illustrated in Figure 2, was built using PHASE [33]. This model describes three-dimensionally all necessary features to block [^3^H]preS1-peptide binding to NTCP. Seven scenarios with different numbers of partial least squares (PLS) factors of the chosen regression model were generated and statistically analyzed (Table 2). A number of four PLS factors revealed the highest value of Pearson-r for the predicted activities of the test set of 0.4614. Therefore, this model composition was chosen for all further investigations. The standard deviation (SD) of the regression for the chosen model was at 0.153 with a coefficient of determination (R^2^) of 0.9519. The stability index of −0.242 of this model illustrates that the texture of the model is strongly dependent on the training set composition. The variance ratio (F) of 357.3 indicates statistically significant regression and the significance level (P) of −41 indicates a great degree of confidence for the variance ratio (Table 2).

Distribution of the atom types of the QSAR model is shown in Table 3 for all seven scenarios of PLS factors. Listed are the percentages of H-bond donor, hydrophobic/non-polar, negative/positive ionic, electron withdrawing, and other regions in the model. The proportions of these attributes point to the relative importance of each attribute for the NTCP inhibitory potency of the respective compound. Interestingly, the attribute proportions did not strongly differ between the respective numbers of PLS factors and were calculated to ~5% H-bond donor, ~60% hydrophobic/non-polar, <1% negative or positive ionic, and ~30% electron withdrawing (Table 3). This means that the amount and distribution of hydrophobic or nonpolar regions is the most important factor for the potency of the inhibitor, while positive or negative ionic residues are of low importance. As expected, there was a strong correlation between the experimentally measured and the QSAR-based predicted binding affinity of the 59 training set compounds with R^2^ of 0.9591 and slope of 0.959 (Figure 3A). Based on this, the binding affinities of the 26 test-set compounds were predicted via the atom-based QSAR model and these ranged from 0.918 (IC_50_ > 1000 µM) to 5.836 (IC_50_ = 1.5 µM) with a mean error of prediction of −0.005 (Table 4). Figure 3B shows the correlation between the experimentally measured and the QSAR-based predicted binding affinities of the 26 test-set compounds that revealed R^2^ of 0.2163 and slope of 0.2439. For five out of the 26 compounds the prediction was quite exact, representing ~20% high-level predictability These compounds are steroid 7s (error −0.01), raloxifene (error 0.057), compound S985852 (error −0.08), steroid 2c (error 0.102), and pioglitazone (error 0.119) (Table 4).

To limit the computing power for the VS with the generated QSAR model, the compounds of the ZINC^15^ library were preselected by screening with an anti-preS1 activity pharmacophore model (Figure 4). All settings for the pharmacophore hypothesis generation are described in the Material and Methods Section 2.7. Active compounds with IC_50_ < 10 µM for inhibition of [^3^H]preS1-peptide binding to NTCP were used to determine features of the pharmacophore and inactive inhibitors with IC_50_ > 20 µM were used to define excluding volumes. As shown in Figure 4, the hypothesis of an anti-preS1 activity pharmacophore model revealed three hydrophobic spheres and one H-bond acceptor sphere together with clustered excluding volumes. Subsequently, ~11 million compounds of the ZINC^15^ library were screened with the illustrated pharmacophore hypothesis. In addition, drug likeness filtering was applied by PHASE [33] for this virtual screen. More than 177,000 hit compounds were identified that matched with all pharmacophore features, representing a hit rate of approximately 1.6%. These compounds then were further screened with the atom-based QSAR model resulting in a compound list with predicted anti-preS1 activities. The top 20 hits that were commercially available are listed in Table 5 and their chemical structures are illustrated in Figure 5. The predicted IC_50_ values for [^3^H]preS1-peptide binding inhibition at NTCP ranged from 7 to 16 µM (Table 5).

These 20 compounds were purchased and tested at 100 µM inhibitory concentrations in binding experiments with 5 nM [^3^H]preS1-peptide and transport experiments with 1 µM [^3^H]TC, both in NTCP-overexpressing HEK293 cells. Among this compound set only the compounds ZINC000012520032, ZINC000253533654, ZINC000253533159, and ZINC000252677946 revealed more than 50% [^3^H]preS1-peptide binding inhibition (Figure 6). Therefore, only these four compounds were further analyzed for IC_50_ inhibitory concentrations (Figure 7) and proof-of-concept HDV infection inhibition (Figure 8). As shown in Figure 7, all four compounds revealed concentration-dependent inhibition of [^3^H]TC transport via NTCP and [^3^H]preS1-peptide binding to NTCP with IC_50_ ranging from 11 to 51 µM and 9 to 35 µM, respectively (Table 6). All compounds were nearly equipotent in both inhibitory assays and, therefore, can be classified as novel non-selective NTCP inhibitors. Of note, the experimentally determined IC_50_ values for [^3^H]preS1-peptide binding inhibition fulfilled quite well the QSAR prediction with a deviation factor of <2.5 for all four compounds. In particular, compound ZINC000253533654 showed almost exactly the predicted activity. Summarizing that from a data set of almost 11 million chemical compounds from the ZINC^15^ library, a subset of 20 compounds could be filtered out, of which four compounds indeed showed concentration-dependent inhibition of myr-preS1_2–48_ lipopeptide binding. So we obtained a predictability value of approximately 20% for our two-step VS approach.

Following the workflow, the four hit compounds were experimentally validated for their inhibitory potency on in vitro HDV infection in NTCP-expressing HepG2 hepatoma cells (Figure 8). All four compounds showed significant concentration-dependent inhibition of HDV infection with a potency rank order of ZINC000253533654 > ZINC000012520032 > ZINC000253533159 > ZINC000252677946. Cytotoxicity studies revealed no toxic effects even at highest inhibitor concentration of 300 µM over 6 h of incubation, representing the experimental conditions of the HDV infection experiments (Figure 9).

## 4. Discussion

The aim of the present study was to identify novel inhibitors of NTCP with drug-like characteristics as potential therapeutics against HBV and HDV infections. HTS was not an option due to financial limitations. In addition, structure-based drug design was not possible, because no valid structural model is currently available for human NTCP. So, we decided to apply ligand-based bioinformatic methods for pharmacophore- and QSAR-guided VS of compound libraries. Our ligand-based approach is not necessarily a disadvantage, because inhibitor design solely based on crystallographic structures of the target protein can be critical due to inadequate resolution [48] or crystallization-related artifacts of the ligand–protein complex [49]. Furthermore, crystallographic studies may ignore discrete conformational states and anisotropic motion of the protein [50,51].

For ligand-based drug design, however, some aspects have to be taken into account to achieve appropriate and valid results. Data from congeneric experiments are necessary, all using the identical target, cell line, and experimental assay [33]. Therefore, we performed inhibition studies for all compounds used for pharmacophore and QSAR modeling with the identical experimental setup. But as NTCP seems to have different substrate and inhibitor binding sites [24], we cannot be sure that all analyzed compounds bind to the identical binding site at NTCP. This is a limitation of the approach used and a possible reason for inaccurate predictions. Also, we cannot exclude that some compounds inhibit NTCP in a competitive manner, while others may induce allosteric effects. These limitations underline the importance of the proof-of-concept in vitro HDV infection experiments that we performed with the four best-performing compounds of the present study.

The design of our inhibition and infection studies allowed us to categorize the identified inhibitors as HDV entry inhibitors acting at NTCP, as (I) their binding to NTCP was demonstrated by inhibition of [^3^H]TC uptake and [^3^H]preS1-peptide binding and (II) they were only present in the infection assay for the first 6 h of HDV exposure of the NTCP-HepG2 cells, representing the early entry phase. However, since we cannot rule out the possibility that some inhibitors might also be transported via NTCP into the HepG2 cells, additional post-entry anti-HDV effects might also be possible.

Starting from ~11 million compounds of the ZINC^15^ library, we identified, in our two-step pharmacophore and QSAR VS, four out of 20 compounds that fulfilled potent [^3^H]preS1-peptide binding inhibition as predicted by the QSAR model and additional proof-of-concept concentration-dependent antiviral activity in the in vitro HDV infection experiments. We thus achieved a predictability of approximately 20% for our VS system, which lays in the acceptable range of 1% to 40% for such approaches [22]. These results indicate that the percentage of compounds that do not optimally fulfil the basic requirements for ligand-based VS is low enough in our data set to obtain a reliable prediction.

When considering the correlation of measured versus predicted activities of the test set of the generated QSAR model, a R^2^ value of 0.2163 seems to be very weak. However, it is recommended not to define the accuracy of a generated QSAR model by its R^2^, due to its sensitivity to the variance in the dependent variable [52]. Furthermore, as the test set was considerably smaller than the training set, fluctuations in the R^2^ value of the test set could easily occur due to fluctuations in the test set variance [52]. The fact that the QSAR model predicted the activity of five test-set compounds out of 26 nearly exactly should be taken into consideration rather than the calculated R^2^. In our VS system, we were able to reproduce this predictability of approximately 20%. In addition, hit compounds not only showed inhibitory potency to block [^3^H]preS1-peptide binding to NTCP but also significantly reduced HDV infection in a concentration-dependent manner. This clearly supports the applicability of our screening system for the discovery of novel HBV/HDV inhibitors acting at NTCP. Of note, none of the hit compounds showed any cytotoxic effects on the HepG2 cell line used for infection studies, even at the highest concentrations of 300 µM. This makes these compounds attractive for further development.

Subsequent studies can be versatile. Obtained data can be used as additional input for recalculations of the generated pharmacophore and QSAR models. In addition, further hits from the top-100 list of predicted [^3^H]preS1-peptide binding inhibitors can be experimentally validated and used for model optimization. Furthermore, the results of the present study can be assessed on the basis of the outcome of our previous studies with the betulin and propanolamine derivatives [13,14]. For both compound groups we could clearly show that only small molecular changes had significant impact on the anti-preS1 activity of the individual compounds. Furthermore, by chemical modifications we achieved a certain virus selectivity of the compounds, which is advantageous to maintaining the physiological bile acid transport function of NTCP during preS1/virus binding inhibition. As an example, the propanolamine compound A000295231 revealed a selectivity index (calculated from the mean IC_50_ for transport inhibition/preS1 binding inhibition) of 65. In the case of the betulin derivatives, 3,28-di-*O*-acetyl-29-hydroxybetulin revealed quite potent inhibition of the [^3^H]preS1-peptide binding to NTCP, but did not inhibit the [^3^H]TC transport via NTCP at all. Based on this, it would be worth generating and test sets of structural derivatives for their anti-preS1 activity. Of note, the compounds ZINC000253533654, ZINC000252677946, and ZINC000253533159 are structural homologs all based on a steroid core structure, the same as for the groups of betulin and arylmethylamino steroids. Based on this, a steroid core structure might be favorable for NTCP inhibitors.

Apart from the goal to find potent and virus-selective NTCP inhibitors for HBV/HDV entry inhibition, potent bile acid transport inhibitors acting on NTCP might also have clinical implications. As examples, patients with cholestatic liver diseases, obesity, dyslipidemia, nonalcoholic steatohepatitis, or primary biliary cholangitis could profit from hepatic bile acid uptake inhibition [10,53]. Based on this, the data of the present study can also be used for pharmacophore and QSAR modelling with a focus on potent bile acid inhibition irrespective of anti-preS1 activity [24,53].

In conclusion, the present study demonstrates, for the first time, pharmacophore and QSAR models for preS1-peptide binding inhibition at NTCP. With a two-step VS approach, novel NTCP inhibitors were identified with high prediction rate and accuracy and even demonstrated anti-HDV activity. These compounds can be used for further development of small molecular HBV/HDV entry inhibitors.

## Figures and Tables

**Figure 1 viruses-13-01489-f001:**
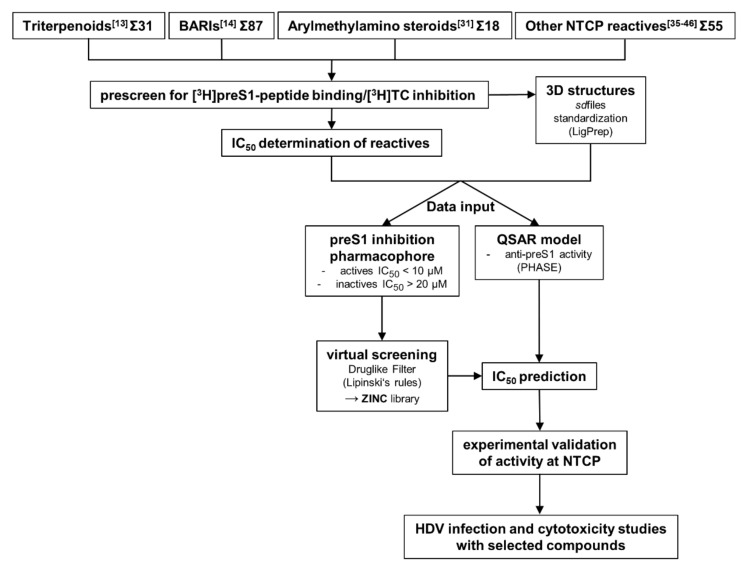
Workflow of the study. The study started with four different compound groups. Three of them were identified as NTCP inhibitors before (see references [13,14,35,36,37,38,39,40,41,42,43,44,45,46]). Arylmethylamino steroids were shown to have antiparasitic activity (see reference [31]) and were established as novel NTCP inhibitors for the first time in the present study. BARIs = bile acid reabsorption inhibitors; QSAR = quantitative structure-activity relationship.

**Figure 2 viruses-13-01489-f002:**
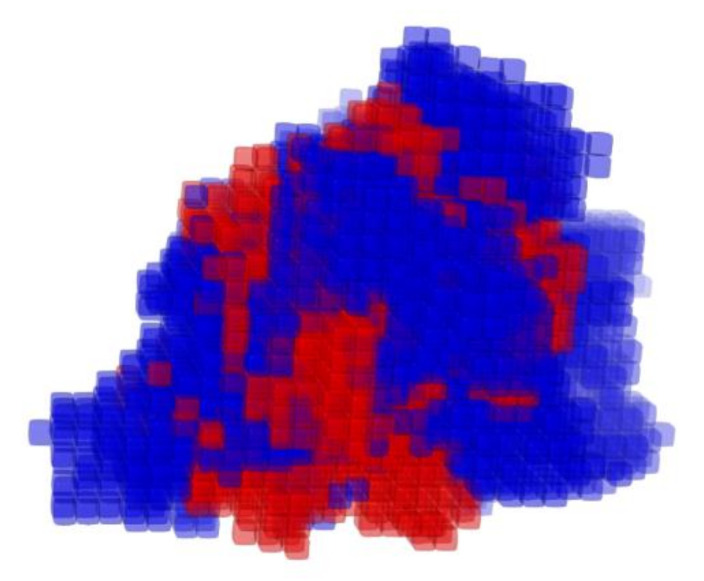
Atom-based QSAR model of anti-preS1 activity. The three dimensional view of the QSAR model illustrates the spatial distribution of contributions to the model. The cubes of the model are colored by the sign of the coefficient: blue for positive coefficients and red for negative coefficients. Positive coefficients indicate an increase in activity, negative coefficients a decrease. Edge length of cubes = 1 Å.

**Figure 3 viruses-13-01489-f003:**
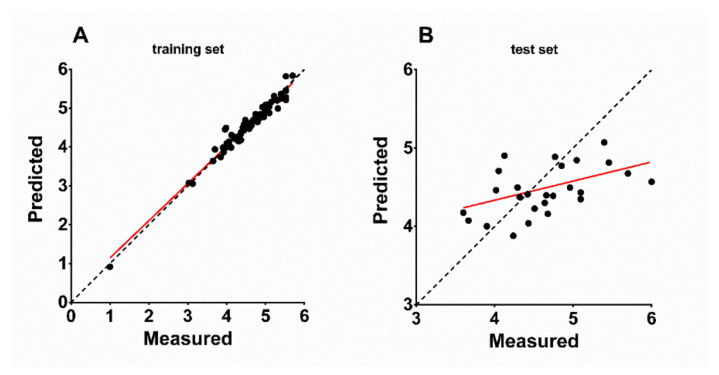
Predicted vs. measured binding affinities (−log (IC_50_ M)) of (**A**) training set and (**B**) test set NTCP inhibitor compounds based on the generated atom-based QSAR model. Red lines indicate linear regression of the scatterplots. (**A**) Training set (59 compounds, Table 1): R^2^ = 0.9591, slope = 0.959. (**B**) Test set (26 compounds, Table 4): R^2^ = 0.2163, slope = 0.2439. Bisector (dashed line) illustrates optimal prediction.

**Figure 4 viruses-13-01489-f004:**
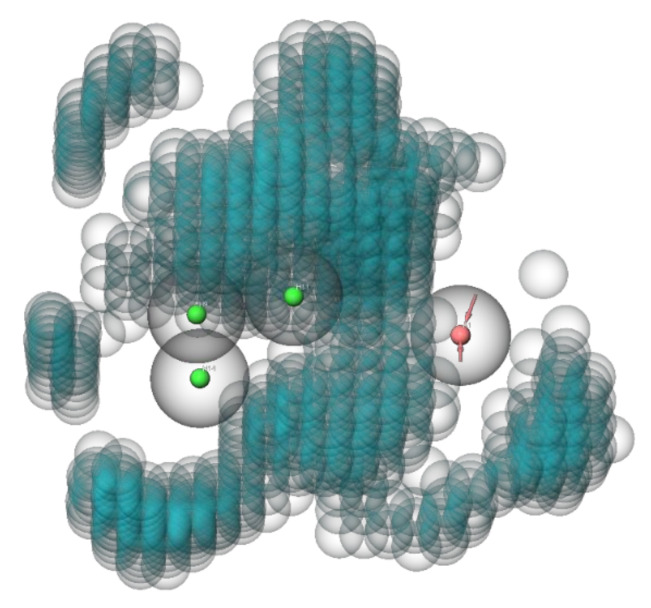
Anti-preS1 activity pharmacophore model for virtual pre-screen of the ZINC^15^ library. The model consists of three hydrophobic spheres (shown with green center) and one H-bond acceptor sphere (shown with red center). Excluding volumes are illustrated as blue bubbles. Binding affinities and structures of ligands from Table 1 and Table 4 served as data input. Excluded volume sphere radii are 1 Å.

**Figure 5 viruses-13-01489-f005:**
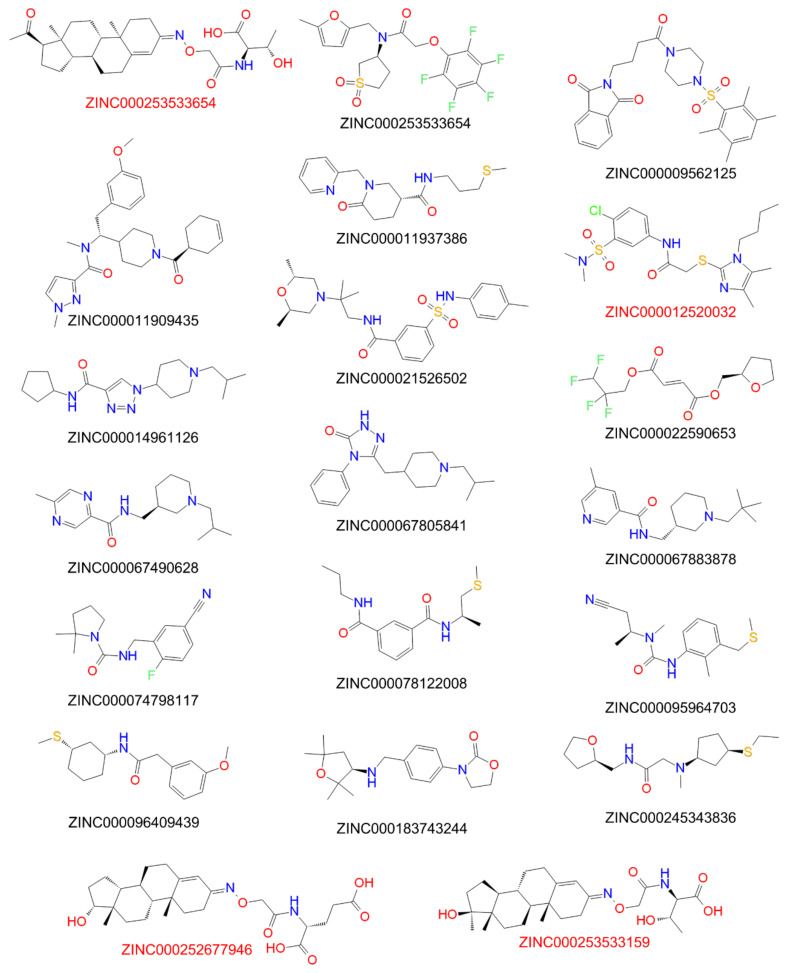
Structures of the top-20-hit compounds from virtual QSAR screening. Red-marked compounds proved anti-HDV activity (see Figure 8).

**Figure 6 viruses-13-01489-f006:**
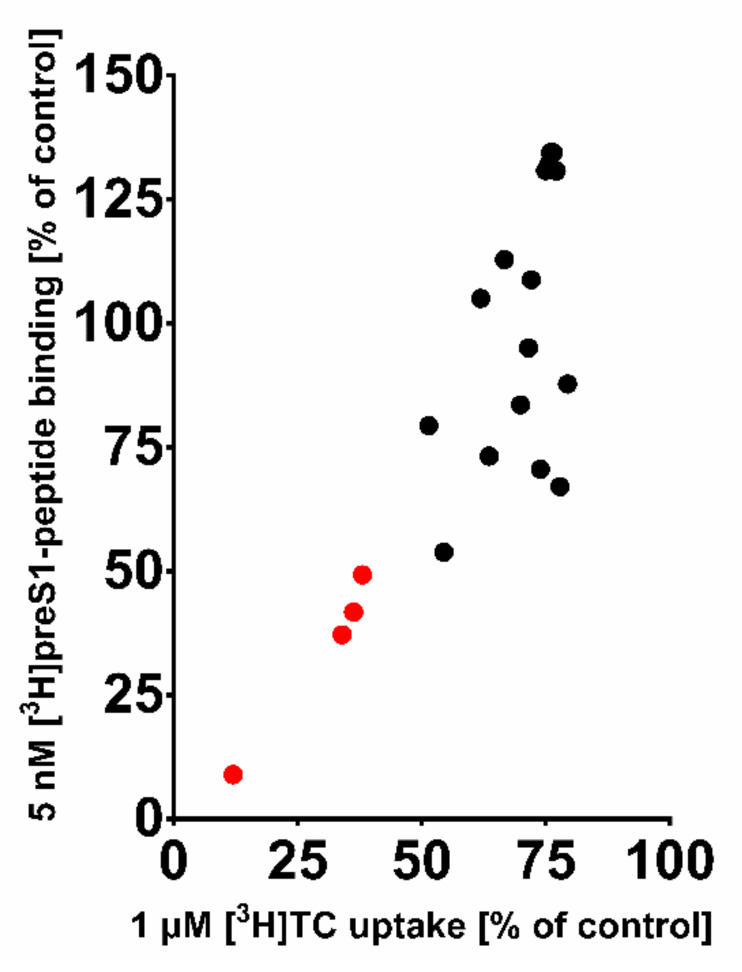
Residual [^3^H]TC transport activity via NTCP vs. residual [^3^H]preS1-peptide binding to NTCP in the presence of the top-20-hit compounds from the virtual QSAR screen. All compounds were used at 100 µM inhibitor concentrations in transport assays with 1 µM [^3^H]TC and binding assays with 5 nM [^3^H]preS1-peptide, both in NTCP-HEK293 cells for 10 min at 37 °C. Compounds resulting in less than 50% residual [^3^H]preS1-peptide binding compared to solvent control are highlighted in red. For these compounds IC_50_ values were determined as indicated in Figure 7.

**Figure 7 viruses-13-01489-f007:**
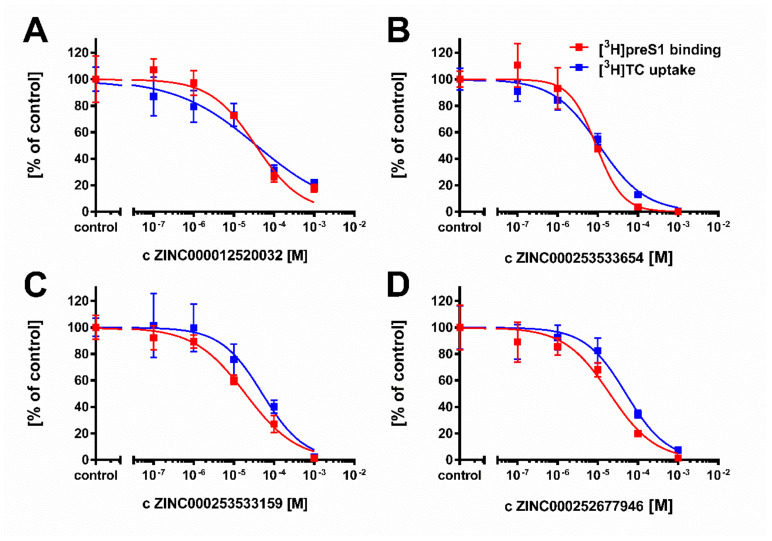
Determination of IC_50_ values of selected compounds for inhibition of [^3^H]preS1 peptide binding to NTCP and [^3^H]TC transport via NTCP at increasing inhibitor concentrations. (**A**) Substance ZINC000012520032, (**B**) Substance ZINC000253533654, (**C**) Substance ZINC000253533159; (**D**) Substance ZINC000252677946. NTCP-HEK293 cells were seeded onto 96-well plates and were incubated with tetracycline to induce expression of NTCP. Cells without tetracycline treatment were used as 0% controls for both assays. Bile acid transport experiments were performed with 1 µM [^3^H]TC and binding experiments were performed with 5 nM [^3^H]preS1-peptide. Both assays were performed over 10 min at 37 °C with increasing concentrations of the indicated inhibitors. Control experiments were performed with solvent alone (set to 100%). The mean of the 0% control was subtracted to calculate net [^3^H]TC transport rates (shown in blue) as well as net [^3^H]preS1 binding rates (shown in red), which are expressed as % of control at the *y*-axis. IC_50_ values were calculated by nonlinear regression analysis using the equation log(inhibitor) vs. response (GraphPad Prism). Data represent means ± SD of quadruplicate determinations of representative experiments.

**Figure 8 viruses-13-01489-f008:**
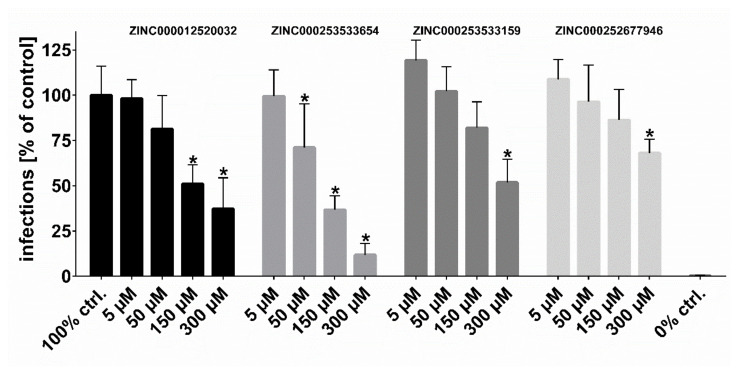
HDV infection studies. NTCP-HepG2 cells were pre-incubated for 5 min with the indicated concentrations of the respective inhibitors in DMEM at 37 °C. Then, cells were additionally inoculated with 120 genome equivalents/cell of HDV particles at 37 °C. After 6 h, cells were washed and further incubated with inhibitor- and virus-free medium, and medium was changed every 3–4 days. At day 9 post infection, cells were fixed and an immunostaining against the HDAg was performed, as a marker of HDV infection. The number of infected cells per well was determined by fluorescence microscopy. NTCP-HepG2 cells incubated without inhibitor were used as control (set to 100% infection rate). Infection experiments in the presence of 0.5 µM myr-preS1_2–48_ lipopeptide served as 0% control (representing 0% infection rate). Data represent means ± SD of three independent experiments, each with triplicate determinations (n = 9). * Significantly lower than 100% control with *p* < 0.01.

**Figure 9 viruses-13-01489-f009:**
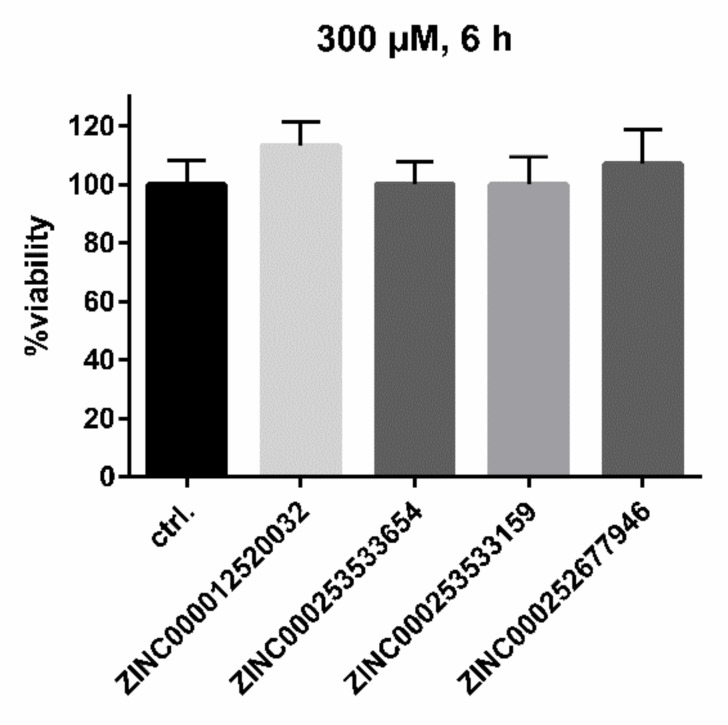
Cytotoxicity studies in NTCP-HepG2 cells. An MTT cytotoxicity assay was performed with the indicated compounds at 5 µM, 50 µM, 150 µM, and 300 µM inhibitor concentrations, incubated over a time span of 6 h at 37 °C. Solvent control was set to 100%. Data represent means ± SD of two independent experiments, each with triplicate determinations (n = 6). Even at the highest concentrations, no cytotoxic effects could be observed.

**Table 1 viruses-13-01489-t001:** Training set to generate an atom-based QSAR model for anti-preS1 activity. Listed are experimentally determined binding affinities (−log (IC_50_ M)) of ligands and predicted binding affinities by the resulting QSAR model. Error describes the difference from the experimentally determined to the predicted binding affinity. Previous descriptions of the compounds as NTCP inhibitors are referenced. As an exception, the arylmethylamino “steroid” compounds are introduced here as novel NTCP inhibitors for the first time.

Ligand Name	Binding Affinity	Prediction	Error
Rosiglitazone [35]	5.699	5.836	0.137
28-*O*-Succinoylbetulin [13]	5.523	5.278	−0.245
3-*O*-Acetylbetulinic acid [13]	5.523	5.214	−0.309
Ginkgolic acid 17:1 [39]	5.523	5.821	0.298
Troglitazone [35]	5.523	5.448	−0.075
Ginkgolic acid 15:1 [39]	5.523	5.461	−0.062
28-*O*-(3,3-dimethylglutaroyl)betulin [13]	5.398	5.365	−0.033
Ritonavir [36]	5.398	5.243	−0.155
20,29-Dihydrobetulonic acid [13]	5.319	4.993	−0.326
3,28-Di-*O*-succinoylbetulin [13]	5.301	5.200	−0.101
3-*O*-Caffeoylbetulin [13]	5.222	5.310	0.088
3,28-Di-*O*-(3,3-dimethylglutaroyl)betulin [13]	5.155	5.163	0.008
Steroid 12c [31]	5.097	5.043	−0.054
28-*O*-(Bromoacetyl)betulin [13]	5.097	4.878	−0.219
S973509 [14]	5.046	5.027	−0.019
A000028897 [14]	5.046	5.090	0.044
A000295231 [14]	5.000	5.004	0.004
Erythrosin B [39]	5.000	5.087	0.087
Betulinaldehyde oxime [13]	4.959	4.848	−0.111
28-(2*H*-Tetrahydropyran-2-yl)betulin [13]	4.854	4.746	−0.108
Allobetulin [13]	4.854	4.843	−0.011
Efavirenz [36]	4.796	4.646	−0.149
Rifampicin [47]	4.745	4.849	0.105
Steroid 9c [31]	4.745	4.718	−0.026
28-*O*-Cinnamoylbetulin [13]	4.721	4.715	−0.006
Rapamycin [41]	4.678	4.683	0.005
Simvastatin [37]	4.638	4.554	−0.084
Cyclosporine A [38,43−46]	4.585	4.472	−0.113
Losartan [37]	4.538	4.620	0.082
Methyl betulinate [13]	4.481	4.692	0.211
Steroid 8c [31]	4.469	4.538	0.070
3-*O*-Acetyl-28-(*2H*-tetrahydropyran-2-yl)betulin [13]	4.456	4.593	0.137
Nimodipine [37]	4.444	4.429	−0.015
Betulonic aldehyde [13]	4.432	4.426	−0.006
Bromosulfophthalein [42]	4.420	4.506	0.085
3-*O*-Acetylbetulin [13]	4.377	4.389	0.012
Steroid 13c [31]	4.367	4.182	−0.184
3,28-Di-*O*-acetyl-20,30-epoxybetulin [13]	4.357	4.321	−0.035
Steroid 1g [31]	4.310	4.148	−0.162
Steroid 1o [31]	4.260	4.255	−0.005
28-*O*-Nicotinoylbetulin [13]	4.201	4.258	0.057
3,28-Di-*O*-acetyl-18,19-dehydro-20,29-dihydrobetulin [13]	4.201	4.235	0.034
Bendroflumethiazide [37]	4.201	4.263	0.062
Tioconazole [37]	4.125	4.308	0.183
S973515 [14]	4.119	3.989	−0.130
Steroid 7c [31]	4.066	4.142	0.076
Steroid 5c [31]	4.056	4.066	0.011
A000295480 [14]	4.009	4.098	0.089
Steroid 3c [31]	3.996	3.994	−0.002
Betulin [13]	3.963	4.454	0.492
Steroid 4c [31]	3.959	4.005	0.046
A000295013 [14]	3.924	3.865	−0.059
Steroid 1e [31]	3.910	3.988	0.078
4′-Ethyl-1′,2′,4′-triazoline-3′,5′-dione-fused 3,28-di-*O*-acetylbetulin [13]	3.848	3.741	−0.107
3,28-Di-*O*-acetylbetulin [13]	3.701	3.939	0.238
Steroid 1c [31]	3.652	3.642	−0.010
Olmesartan [37]	3.129	3.057	−0.072
Irbesartan [37]	3.032	3.076	0.045
Rosuvastatin [37]	1.000	0.918	−0.082

**Table 2 viruses-13-01489-t002:** Statistics of the generated atom-based QSAR model for anti-preS1 activity. Listed are data of a total of seven scenarios with different numbers of partial least squares factors of the regression model (# PLS Factors). SD = standard deviation of the regression; R^2^ = coefficient of determination for the regression of the scatter plot of the training set (see Table 1 and Figure 3A); stability = stability of the model predictions to changes in the training set composition with a maximum value of 1 (meaning stable); F = variance ratio (large values of F indicate a more statistically significant regression); P = significance level of variance ratio (smaller values indicate a greater degree of confidence); Pearson-r = value of Pearson-R for the predicted activities of the test set (see Table 4 and Figure 3B). A number of four PLS factors revealed the highest Pearson-r for the prediction of activities of the training set ligands (0.4614) and therefore was chosen as the model for the first VS.

# PLS Factors	SD	R^2^	Stability	F	P	Pearson-r
1	0.4671	0.5998	0.245	95.9	−14	0.2709
2	0.3093	0.8273	0.055	150.9	−25	0.3074
3	0.2314	0.9048	−0.216	196.5	−31	0.358
4	0.153	0.9591	−0.242	357.3	−41	0.4614
5	0.1038	0.9815	−0.298	635.2	−50	0.408
6	0.079	0.9894	−0.335	922	−56	0.4073
7	0.053	0.9953	−0.345	1766.9	−65	0.4124

**Table 3 viruses-13-01489-t003:** Distribution of atom types of the generated QSAR model. Listed are numbers of partial least squares factors of the regression model (# PLS factors), the relative proportions to the model of H-bond donor, hydrophobic (= non-polar), negative ionic, positive ionic, electron-withdrawing, and other atom types. The data for four PLS factors are highlighted since this model was used for VS.

# PLS Factors	H-Bond Donor	Hydrophobic/non-Polar	Negative Ionic	Positive Ionic	Electron-Withdrawing	Other
1	0.045	0.635	0	0	0.285	0.035
2	0.048	0.618	0	0	0.297	0.037
3	0.049	0.613	0	0	0.300	0.038
4	0.053	0.606	0	0	0.301	0.040
5	0.055	0.601	0	0	0.301	0.041
6	0.056	0.600	0.001	0.001	0.302	0.041
7	0.056	0.601	0	0	0.302	0.040

**Table 4 viruses-13-01489-t004:** Test set to validate the atom-based QSAR model for anti-preS1 activity. Listed are experimentally determined binding affinities (−log (IC_50_ M)) of ligands and predicted binding affinity by the resulting QSAR model. Error describes the difference between the experimentally determined and the predicted binding affinity. Previous descriptions of the compounds as NTCP inhibitors are referenced. As an exception, the arylmethylamino “steroid” compounds are introduced here as novel NTCP inhibitors for the first time.

Ligand Name	Binding Affinity	Prediction	Error
Ciglitazone [35]	6.000	4.570	−1.430
Betulinic acid [13]	5.699	4.677	−1.022
20,29-Dihydrobetulin [13]	5.456	4.815	−0.641
3-*O*-(3,3-Dimethylglutaroyl)betulinic acid [13]	5.398	5.073	−0.325
3,28-Di-*O*-acetyl-29-hydroxybetulin [13]	5.097	4.433	−0.664
Saquinavir [36]	5.097	4.349	−0.748
Emodin [39]	5.046	4.845	−0.201
Ginkgolic acid 13:0 [39]	4.959	4.496	−0.462
S985852 [14]	4.854	4.774	−0.080
Pioglitazone [35]	4.770	4.888	0.119
Nitrendipine [37]	4.745	4.390	−0.354
Glyburide [37]	4.678	4.161	−0.517
Lupenone [13]	4.658	4.398	−0.259
Betulonoyl dimethyl-L-aspartate [13]	4.638	4.300	−0.338
3-Oxoallobetulin [13]	4.509	4.228	−0.281
Steroid 6c [31]	4.432	4.040	−0.392
Steroid 7s [31]	4.420	4.410	−0.010
Raloxifene [37]	4.319	4.376	0.057
Lupeol [37]	4.292	4.497	0.204
Steroid 1s [31]	4.237	3.881	−0.356
3,28-Di-*O*-(dihydrocinnamoyl)betulin [13]	4.125	4.902	0.777
A000289041 [14]	4.051	4.708	0.657
Nifedipine [37]	4.018	4.463	0.445
Steroid 2c [31]	3.900	4.002	0.102
Steroid 2s [31]	3.666	4.075	0.410
Steroid 2o [31]	3.600	4.176	0.576

**Table 5 viruses-13-01489-t005:** Top 20 commercially available hits from VS of the ZINC^15^ compound library [20] (total amount of compounds: 10,844,830; downloaded on 15 May 2020; reactivity: standard; purchasability: in-stock; pH: ref mid (7.4); charge: −2, −1, 0, +1, +2; subset: drug-like). All downloaded data were processed as described in the Material and Methods section. Compounds showing anti-HDV activity are highlighted in red.

Compound ID	Predicted Anti-PreS1 Activity (IC_50_ µM)	Compound ID	Predicted Anti-PreS1 Activity (IC_50_ µM)
ZINC000021526502	7	ZINC000014961126	9
ZINC000253533654	7	ZINC000009431397	9
ZINC000067883878	8	ZINC000096409439	9
ZINC000009562125	8	ZINC000245343836	9
ZINC000253533159	8	ZINC000095964703	10
ZINC000183743244	8	ZINC000022590653	10
ZINC000011937386	8	ZINC000074798117	10
ZINC000067805841	9	ZINC000011909435	11
ZINC000067490628	9	ZINC000252677946	13
ZINC000078122008	9	ZINC000012520032	16

**Table 6 viruses-13-01489-t006:** Predictivity validation of the atom-based QSAR model.

Compound ID	Predicted Anti-PreS1 Activity (IC_50_ µM)	Experimentally Determined Anti-PreS1 Activity (IC_50_ µM)	Experimentally Determined Anti-TC Activity (IC_50_ µM)
ZINC000012520032	16	35	37
ZINC000253533654	7	9	11
ZINC000253533159	8	19	51
ZINC000252677946	13	20	51

## Data Availability

Obtained and analyzed data of this study are available from the corresponding author on request.

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
