# Peer review of "Identification of Novel HBV/HDV Entry Inhibitors by Pharmacophore- and QSAR-Guided Virtual Screening"

_viruses, 2021, doi:10.3390/v13081489_

Round 1

Reviewer 1 Report

Kirstgen et al. in the current manuscript performed a ligand-based bioinformatic approach to generate and validate appropriate pharmacophore and QSAR models as a continued effort towards identification of small molecules NTCP inhibitors against HBV and HDV infection. In previous studies, the group recently identified individual compounds from two different classes (betulin and propanolamine derivatives) that showed myr-preS12-48 lipopeptide binding inhibition and blocked in vitro HDV infection of NTCP-expressing HepG2 cells [Kirstgen, M et al. (2020) Sci. Rep; Kirstgen, M et al. (2021). Viruses]. In the present study, for QSAR modeling, they used IC50 values for the betulin and propanolamine derivatives from their previous studies. While for arylmethylamino steroids, they included new IC50 data presented in this manuscript in Supplementary Figure S1.  The generated pharmacophore and QSAR models were used for virtual screening of drug-like chemicals from the ZINC database (~11 million compounds). The 20 best performing compounds then were experimentally tested for preS1-peptide binding inhibition in NTCP-35 HEK293 cells. Unfortunately, all compounds identified in the study had very low potency for [3H]preS1-peptide binding inhibition ranging from 8 – 251 μM compared to previously reported NTCP inhibitors. At these concentrations, compounds can lead to unspecific binding and inhibition as well as high toxicity after prolonged treatment. The team attempted to address that these compounds are not cytotoxic at these concentrations after incubating them for 6 h and then replacing with inhibitor-free culture medium for additional 24h but has not shown the effect of compounds on cell toxicity for the duration of HDV infection assay.

Recommended experiments:

  1. Provide additional toxicity assays for the duration of HDV infection.

Citations:

Kirstgen, M.; Lowjaga, K.; Müller, S.F.; Goldmann, N.; Lehmann, F.; Alakurtti, S.; Yli-Kauhaluoma, J.; Glebe, D.; Geyer, J. Selec-558 tive hepatitis B and D virus entry inhibitors from the group of pentacyclic lupane-type betulin-derived triterpenoids. Sci. Rep. 559 2020, 10, 21772, doi:10.1038/s41598-020-78618-2. 560

Kirstgen, M.; Lowjaga, K.A.A.T.; Müller, S.F.; Goldmann, N.; Lehmann, F.; Glebe, D.; Baringhaus, K.-H.; Geyer, J. Hepatitis D 561 Virus Entry Inhibitors Based on Repurposing Intestinal Bile Acid Reabsorption Inhibitors. Viruses 2021, 13, 562 doi:10.3390/v13040666

Author Response

Point-by-point response to the reviewer comments

Reviewer #1

Kirstgen et al. in the current manuscript performed a ligand-based bioinformatic approach to generate and validate appropriate pharmacophore and QSAR models as a continued effort towards identification of small molecules NTCP inhibitors against HBV and HDV infection. In previous studies, the group recently identified individual compounds from two different classes (betulin and propanolamine derivatives) that showed myr-preS12-48 lipopeptide binding inhibition and blocked in vitro HDV infection of NTCP-expressing HepG2 cells [Kirstgen, M et al. (2020) Sci. Rep; Kirstgen, M et al. (2021). Viruses]. In the present study, for QSAR modeling, they used IC50 values for the betulin and propanolamine derivatives from their previous studies. While for arylmethylamino steroids, they included new IC50 data presented in this manuscript in Supplementary Figure S1.  The generated pharmacophore and QSAR models were used for virtual screening of drug-like chemicals from the ZINC database (~11 million compounds). The 20 best performing compounds then were experimentally tested for preS1-peptide binding inhibition in NTCP-35 HEK293 cells. Unfortunately, all compounds identified in the study had very low potency for [3H]preS1-peptide binding inhibition ranging from 8 – 251 μM compared to previously reported NTCP inhibitors. At these concentrations, compounds can lead to unspecific binding and inhibition as well as high toxicity after prolonged treatment. The team attempted to address that these compounds are not cytotoxic at these concentrations after incubating them for 6 h and then replacing with inhibitor-free culture medium for additional 24h but has not shown the effect of compounds on cell toxicity for the duration of HDV infection assay.

Recommended experiments: 1. Provide additional toxicity assays for the duration of HDV infection.

Response: We agree with the reviewer that cytotoxicity of the compounds is a relevant and very important issue. We addresses this point with cytotoxicity studies (HepG2 cells + inhibitor over 6 h) that were designed according to the in vitro HDV infection experiments (HepG2 cells + HDV + inhibitor over 6 h). In both experimental setups the inhibitor compounds were added to the cells over 6 h. Thereafter, the cells are incubated without inhibitor and without virus in the medium. Therefore, we are convinced that our cytotoxicity studies over 6 h with the highest inhibitor concentration of 300 µM are valid and sufficient to address the question if the effects on infection may result from cytotoxicity to the cells and not from virus uptake inhibition. During this 6 h period we expect attachment of the virus particles to NTCP and endocytosis of the virus-receptor complex. Presence of inhibitor during this phase blocks virus binding to its receptor and thereby entry of the virus-receptor complex. In our point of few this is the only possible and reasonable setup for cytotoxicity studies. Of note, after this 6 h period, the NTCP-HepG2 cells are further cultivated in inhibitor- and virus-free medium for 9 additional days. During that time that the virus replicates and the cells produce the HDV antigen that we then detected as robust marker of productive virus infection. We also discussed different other setups for cytotoxicity studies in the study planning phase including incubation of the cells over the whole 9 days with inhibitor or measurement of the cytotoxicity in parallel to the HDV antigen detection after 9 days. However, both scenarios do not reflect the real experimental setup or cytotoxicity measurements in parallel to HDV antigen detection could be influenced by the degree of virus entry and infection and, therefore, would have no value for the cytotoxicity determination of the inhibitor compounds per se. Of note, the readout of our infection assays are numbers of infected cells (HDV antigen positive cells) at a certain endpoint. These cells were manually counted and all wells underwent quality control for morphologically intact cells at this endpoint. Due to the fact that HDV is not lytic we generally did not observe any cell loss over time. Nevertheless, all wells were checked for cell loss and atypical morphology at the endpoint. Here we could not find any differences between the treated or untreated cells. The identical cytotoxicity testing approach was also used in our previous publications (Kirstgen et al. (2020) Sci. Rep, Kirstgen et al. (2021) Viruses) in exactly the same way.

Reviewer 2 Report

The authors had made an excellent effort in setting up a QSAR Pharmacophore method for virtually screening large libraries of compounds on HBV/HDV antiviral activity but this could be used in more case studies. Although the effort given some points arises that needs to be addressed.

Line specific remarks:

Line 141: Abbreviation HGM is explained in line 144 while 141 is the first mention of it.

Lines 189-200: Authors should provide more detailed information if possible. Please check general remark 2.

Line 248: Compound 12c is highlighted from the citing literature, hence it would be helpful to readers a mention of the IUPAC name or a figure with the chemical structure or both.

Line 251: SDF files are compiled in 2D or 3D format? Please clarify accordingly.

Lines 253-257: The authors use four categories of compounds based on references Group I [13], Group II [14], Group III [29], Group IV [33-44]. It would be more comprehensive for the readers to know the sum of each group of compounds, as the authors clearly state Group I and II being 31 and 87 respectively this should be stated also for the rest groups. This is due to the fact that in text the transition on how and from which groups the 85 compounds are compiled and from what total. In continuation to this come next lines remarks.

Lines 270-324: In accordance to previous remark it’s not understandable when reading how the classification of training set and test set was performed. Could this be the reason of introducing bias? Maybe the randomness mentioned in section 2.8. is not that random? Plots doesn’t justify it (check general remark 3). Please elaborate.

Lines 333-334: The line refers to method section for more information but the section itself is very frugal; please revise accordingly to line remarks ….

Line 356: How come the authors included charged compounds since in the Pharmacophore setting step it shows no contribution? At this point authors could also introduce something relevant to general remark 6, if not elsewhere.

General remarks:

  1. Can the authors mention in text an approximate percentage of NTCP overexpression with the tetracycline/doxycycline usage?
  2. The filtering options are not clearly given; what features have been used in PHASE? Not referring to settings which are provided. Did the authors include any user defined (custom) features?
  3. Indeed as the authors state on the conclusions the rationalization of the structure activity data for test group is an issue. The number of compounds is marginal but is this only the case or method may include a bias not visible at once?
  4. Why the authors didn’t apply any PAINS filtering?
  5. Why the authors didn’t perform a general antiviral assay on the compounds since as they mention on conclusion section activity may be due to direct HBV/HDV activity?
  6. Subset of ZINC used is Drug-Like but no version is provided (i.e. ZINC12, ZINC15, ZINC20) nor a respective citing literature throughout the text.

Author Response

Point-by-point response to the reviewer comments

Reviewer #2

The authors had made an excellent effort in setting up a QSAR Pharmacophore method for virtually screening large libraries of compounds on HBV/HDV antiviral activity but this could be used in more case studies. Although the effort given some points arises that needs to be addressed.

Line specific remarks: Line 141: Abbreviation HGM is explained in line 144 while 141 is the first mention of it.

Response: We explained the abbreviation HGM in line 141 and adapted the text as follows (lines 140-150): “NTCP-HepG2 cells were pre-incubated for 5 min with inhibitors solved in 80 µl Hepatocyte Growth Medium (HGM) per well in concentrations ranging from 5 µM to 300 µM. Infection experiments were performed in NTCP-HepG2 cells as described [13]. Briefly, during infection, cells were cultured in 96-well plates in HGM consisting of William’s E Medium (Thermo Fisher Scientific) containing …”

Lines 189-200: Authors should provide more detailed information if possible. Please check general remark 2.

Response: We have now included for each software setting whether it is “default setting” or ”custom setting”. Custom settings then are further specified, see lines 184 ff.

Line 248: Compound 12c is highlighted from the citing literature, hence it would be helpful to readers a mention of the IUPAC name or a figure with the chemical structure or both.

Response: We agree with the reviewer, that the chemical structure of the mentioned compound is an important information for the reader. Therefore we added a new Supplementary Figure S2 with structures of the three most potent inhibitors of [3H]preS1 peptide binding inhibition with the following figure legend: “Supplementary Figure S2. Structures of selected arylmethylamino steroids with highest potency of [3H]preS1 peptide binding inhibition. All structures can be found in the publication by Krieg et al. 2017 [30].” Furthermore, we added the following information to the main text (lines 264ff): “The most potent compound was steroid 12c with an IC50 for [3H]preS1-peptide binding inhibition of 8 µM. Structures of the three most potent [3H]preS1-peptide binding inhibitors of this compound class are depicted in Supplementary Figure S2.”

Line 251: SDF files are compiled in 2D or 3D format? Please clarify accordingly.

Response: We added this information as follows: “In addition, data of the chemical structures were collected and were saved as 2D sdfiles.” As described in the Material and Methods section, chapter “2.6 Data preparation”, all imported chemical 2D structures were then standardized using LigPrep.

Lines 253-257: The authors use four categories of compounds based on references Group I [13], Group II [14], Group III [29], Group IV [33-44]. It would be more comprehensive for the readers to know the sum of each group of compounds, as the authors clearly state Group I and II being 31 and 87 respectively this should be stated also for the rest groups. This is due to the fact that in text the transition on how and from which groups the 85 compounds are compiled and from what total. In continuation to this come next lines remarks.

Response: We added the missing numbers and now write: “These experiments were performed with four different groups of test compounds, including 31 betulin-derived triterpenoids [13], and 87 propanolamine derivatives [14]. Both groups of compounds were analyzed before as novel NTCP inhibitors in our laboratory. In addition, a set of 18 arylmethylamino steroids that previously showed antiparasitic activity against Plasmodium falciparum and Schistosoma mansoni [31] was used for NTCP inhibition for the first time in the present study (Supplementary Figures S1 and S2). And finally, a group of 55 structurally unrelated compounds that were reported as NTCP inhibitors in the literature [35-46] were included in the [3H]preS1-peptide binding inhibition experiments.“ In addition, the number of compounds tested for each compound group were included in Figure 1.

Lines 270-324: In accordance to previous remark it’s not understandable when reading how the classification of training set and test set was performed. Could this be the reason of introducing bias? Maybe the randomness mentioned in section 2.8. is not that random? Plots doesn’t justify it (check general remark 3). Please elaborate.

Response: We comprehend this remark of the reviewer and reworked this paragraph for better clarity as follows (lines 253ff): “However, for some of the compounds a sigmoidal concentration-dependent dose-response relationship could not be determined experimentally. Therefore, these compounds were removed from the dataset. This was important to avoid distortion of subsequent models due to non-valid IC50 data. In total, a data set consisting of 85 compounds with valid IC50 data could be compiled.” Additionally lines 272ff were improved: “In total, the IC50 values for [3H]preS1-peptide binding inhibition from the 85 data set compounds revealed binding affinities between 1 (IC50 > 1,000 µM) and 5.699 (IC50 = 2 µM). The compounds then were divided into a training set to generate the QSAR model (59 compounds, Table 1) and a test set to validate the QSAR model (26 compounds, Table 4) as described in the Material and Methods section 2.8.”

Lines 333-334: The line refers to method section for more information but the section itself is very frugal; please revise accordingly to line remarks ….

Response: We specified this to: “All settings for the pharmacophore hypothesis generation are described in the Material and Methods section 2.7.“

Line 356: How come the authors included charged compounds since in the Pharmacophore setting step it shows no contribution? At this point authors could also introduce something relevant to general remark 6, if not elsewhere.

Response: We thank the reviewer for this interesting remark. We discussed this point intensively and came to the conclusion that we would interpret the pharmacophore model somewhat different. Just because there is no specific attribute for a charge in the common feature pharmacophore, one cannot exclude this possibility. It just implies that the substances from which the pharmacophore model was generated have no significant conformity in terms of a charge. As long as the presence of a charge does not collide with the excluding values, we believe it is generally permitted. When looking at the QSAR model, it is noticeable that the presence of a charge is not attributed any significant role here either (compare to Table 3). However, in our publication about betulin derivatives (reference 13), we were able to demonstrate that the presence of a negative charge appears to have a major influence on the interaction of inhibitors with the binding domain of taurocholic acid at NTCP. Negatively charged inhibitors also showed strong inhibitory effects on preS1 peptide binding in this study. However, loss of this negative charge did not negatively affect inhibitory potency in the case of preS1 peptide binding inhibition. Only the inhibitory effect on bile acid transport was greatly reduced by the loss of the negative charge. We therefore consider it useful to also include charged substances in our screening system.

General remarks:

  1. Can the authors mention in text an approximate percentage of NTCP overexpression with the tetracycline/doxycycline usage?

Response: As both cells lines used, HEK293 and HepG2, lack any endogenous NTCP expression, there is no good basis for such a calculation. This question was intensively analyzed and discussed in two of our previous publications that are cited here: NTCP-HepG2 in König et al. 2014 (Reference 9) and NTCP-HEK293 in Grosser et al. 2021 (Reference 24) (see Material and Methods section 2.1).

  1. The filtering options are not clearly given; what features have been used in PHASE? Not referring to settings which are provided. Did the authors include any user defined (custom) features?

Response: As already mentioned above, we now indicated for each software setting whether it is a “default setting” or a “custom setting”.

  1. Indeed as the authors state on the conclusions the rationalization of the structure activity data for test group is an issue. The number of compounds is marginal but is this only the case or method may include a bias not visible at once?

Response: One could expect bias at this step of our studies, but it should be considered as a beneficial step for the quality of the resulting models. Some compounds revealed inhibitory potency when they were first tested at 100 µM concentration, but subsequent assays for IC50 determination often revealed invalid inhibition patterns. For example, some compounds revealed a residual [3H]preS1 peptide binding activity of approximately 40 % regardless of the concentration used. In further studies we would like to analyze the mode of action of those interesting compounds. It could be some kind of allosteric effect on NTCP, but this is just a hypothesis. With respect to the studies described in this manuscript using pharmacophore and QSAR models, it was important for us to follow best practice and therefore we consistently used congeneric data. Consequently, it was important for us to include only those substances that produce clear concentration-dependent effects at NTCP. For these compounds, we consider it most likely that they compete competitively with binding of the viral preS1 domain of the large surface protein and therefore demonstrate entry inhibitors of HBV and HDV. The number of compounds was sufficient to generate pharmacophore and QSAR models that together were able to predict activity of new antivirals. Nevertheless, we agree with the reviewer to further expand our dataset in subsequent studies and recalculate our models accordingly. We also think that more data input should be beneficial for model quality and predictivity.

  1. Why the authors didn’t apply any PAINS filtering?

Response: We thank the reviewer very much for this helpful hint. After search at the ZINC15 website, we found that under the “Reactivity” tab, the attribute “Anodyne” must be selected to apply a PAINS filter to the database. The software from SCHRÖDINGER that we use for our studies can apparently also provide a token that enables PAINS filtering. We will definitely take this point into account in future screening approaches.

  1. Why the authors didn’t perform a general antiviral assay on the compounds since as they mention on conclusion section activity may be due to direct HBV/HDV activity?

Response: First step on the way to the studies described in this manuscript was to establish an assay to generate activity data of compounds for our research subject with a relatively high throughput. Important for us was to simultaneously check influence of inhibitors on the virus receptor function and bile acid transporter function of NTCP. Our focus was particularly on the extent to which preS1 peptide binding ability is influenced. For a comparative analysis of this effect with the bile acid transport function, we chose this setup. Infection studies served as verification experiments for the possibility to block especially the entry step of the virus. We needed approximately 3 years to collect the displayed data. To perform infection studies as described in this manuscript for all compounds, we would have needed much more time due to the effort involved in these studies. In addition, based only on the infection experiments, we would not have sufficient evidence that specifically the binding of the viral preS1 domain is inhibited. For the generation of computer models, data from short inhibition experiments, such as the IC50 studies described here, are useful. Use of inhibitors a priori for infection studies would also filter out compounds that block post-entry steps of virus infection. This was out of the scope of our study.

  1. Subset of ZINC used is Drug-Like but no version is provided (i.e. ZINC12, ZINC15, ZINC20) nor a respective citing literature throughout the text.

Response: We changed the word “ZINC” to “ZINC15” throughout the manuscript and added information to the method section as follows: “A set of 20 compounds, which can be found in the ZINC15 database [20] (https://zinc.docking.org/), was purchased from MolPort (Riga, Latvia).”

Round 2

Reviewer 2 Report

Dear Editor,

The authors addressed adequately every point raised in my review.

Moreover, I am sutisfied